# Explain Your Move:
# Understanding Agent Actions Using Specific and Relevant Feature Attribution

**Nikaash Puri**[*,‡]  **Sukriti Verma**[*,‡]  **Piyush Gupta**[*,‡]  **Dhruv Kayastha**[†,§]
**Shripad Deshmukh**[†,¶]  **Balaji Krishnamurthy**[‡]  **Sameer Singh**[‖]

[‡] Media and Data Science Research, Adobe Systems Inc., Noida, Uttar Pradesh, India 201301
[§] Indian Institute of Technology Kharagpur, West Bengal, India 721302
[¶] Indian Institute of Technology Madras, Chennai, India 600036
[‖] Department of Computer Science, University of California, Irvine, California, USA
{nikpuri, sukrverm, piygupta}@adobe.com

## Abstract

As deep reinforcement learning (RL) is applied to more tasks, there is a need to visualize and understand the behavior of learned agents. Saliency maps explain agent behavior by highlighting the features of the input state that are most relevant for the agent in taking an action. Existing perturbation-based approaches to compute saliency often highlight regions of the input that are not relevant to the action taken by the agent. Our proposed approach, SARFA (Specific and Relevant Feature Attribution), generates more focused saliency maps by balancing two aspects (specificity and relevance) that capture different desiderata of saliency. The first captures the impact of perturbation on the relative expected reward of the action to be explained. The second downweighs irrelevant features that alter the relative expected rewards of actions other than the action to be explained. We compare SARFA with existing approaches on agents trained to play board games (Chess and Go) and Atari games (Breakout, Pong and Space Invaders). We show through illustrative examples (Chess, Atari, Go), human studies (Chess), and automated evaluation methods (Chess) that SARFA generates saliency maps that are more interpretable for humans than existing approaches. For the code release and demo videos, see https://nikaashpuri.github.io/sarfa-saliency/.

## 1 Introduction

Deep learning has achieved success in various domains such as image classification (He et al., 2016; Krizhevsky et al., 2012), machine translation (Mikolov et al., 2010), image captioning (Karpathy et al., 2015), and deep Reinforcement Learning (RL) (Mnih et al., 2015; Silver et al., 2017). To explain and interpret the predictions made by these complex, "black-box"-like systems, various gradient and perturbation techniques have been introduced for image classification (Simonyan et al., 2013; Zeiler & Fergus, 2014; Fong & Vedaldi, 2017) and deep sequential models (Karpathy et al., 2015). However, interpretability for RL-based agents has received significantly less attention. Interpreting the strategies learned by RL agents can help users better understand the problem that the agent is trained to solve. For instance, interpreting the actions of a chess-playing agent in a position could provide useful information about aspects of the position. Interpretation of RL agents is also an important step before deploying such models to solve real-world problems.

Inspired by the popularity and use of saliency maps to interpret in computer vision, a number of existing approaches have proposed similar methods for reinforcement learning-based agents. Greydanus et al. (2018) derive saliency maps that explain RL agent behavior by applying a Gaussian blur to different parts of the input image. They generate saliency maps using differences in the value

---

[*]These authors contributed equally
[†]Work done during the Adobe MDSR Research Internship Program.

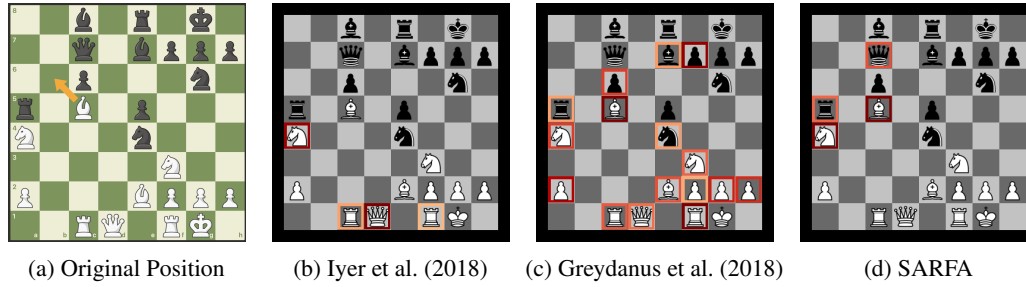

| (a) Original Position | (b) Iyer et al. (2018) | (c) Greydanus et al. (2018) | (d) SARFA |

Figure 1: Saliency maps generated by existing approaches

function and policy vector between the original and perturbed state. They achieve promising results on agents trained to play Atari games. Iyer et al. (2018) compute saliency maps using a difference in the action-value ($Q(s, a)$) between the original and perturbed state.

There are two primary limitations to these approaches. The first is that they highlight features whose perturbation affects actions apart from the one we are explaining. This is illustrated in Figure 1, which shows a chess position (it is white's turn). Stockfish[1] plays the move Bb6 in this position, which traps the black rook (a5) and queen (c7)[2]. The knight protects the white bishop on a4, and hence the move works. In this position, if we consider the saliency of the white queen (square d1), then it is apparent that the queen is not involved in the tactic and hence the saliency should be low. However, perturbing the state (by removing the queen) leads to a state with substantially different values for $Q(s, a)$ and $V(s)$. Therefore, existing approaches (Greydanus et al., 2018; Iyer et al., 2018) mark the queen as salient. The second limitation is that they highlight features that are not relevant to the action to be explained. In Figure 1c, perturbing the state by removing the black pawn on c6 alters the expected reward for actions other than the one to be explained. Therefore, it alters the policy vector and is marked salient. However, the pawn is not relevant to explain the move played in the position (Bb6).

In this work we propose SARFA, *Specific and Relevant Feature Attribution*, a perturbation based approach for generating saliency maps for black-box agents that builds on two desired properties of action-focused saliency. The first, *specificity*, captures the impact of perturbation *only* on the Q-value of the action to be explained. In the above example, this term downweighs features such as the white queen that impact the expected reward of all actions equally. The second, *relevance*, downweighs irrelevant features that alter the expected rewards of actions other than the action to be explained. It removes features such as the black pawn on c6 that increase the expected reward of other actions (in this case, Bb4). By combining these aspects, we generate a saliency map that highlights features of the input state that are relevant for the action to be explained. Figure 1 illustrates how the saliency map generated by SARFA only highlights pieces relevant to the move, unlike existing approaches.

We use our approach, SARFA to explain the actions taken by agents for board games (Chess and Go), and for Atari games (Breakout, Pong and Space Invaders). Using a number of illustrative examples, we show that SARFA obtains more focused and accurate interpretations for all of these setups when compared to Greydanus et al. (2018) and Iyer et al. (2018). We also demonstrate that SARFA is more effective in identifying important pieces in chess puzzles, and further, in aiding skilled chess players to solve chess puzzles (improves accuracy of solving them by nearly 25% and reduces the time taken by 31% over existing approaches).

## 2 SPECIFIC AND RELEVANT FEATURE ATTRIBUTION (SARFA)

We are given an agent $M$, operating on a state space $\mathcal{S}$, with the set of actions $\mathcal{A}_s$ for $s \in \mathcal{S}$, and a $Q$-value function denoted as $Q(s, a)$ for $s \in \mathcal{S}$, $a \in \mathcal{A}_s$. Following a greedy policy, let the action that was selected by the agent at state $s$ be $\hat{a}$, i.e. $\hat{a} = \arg \max_a Q(s, a)$. The states are parameterized in terms of state-features $\mathcal{F}$. For instance, in a board game such as chess, the features are the 64 squares.

---

[1]https://stockfishchess.org/

[2]We follow the coordinate naming convention where columns are 'a-h' (left-right), rows '8-1' (top-bottom), and pieces are labeled using the first letter of its name in upper case (e.g. 'B' denotes the bishop). A move consists of the piece and the position it moves to, e.g. 'Bb6' indicates that the bishop moves to position 'b6'.

For Atari games, the features are pixels. We are interested in identifying which features of the state $s$ are important for the agent in taking action $\hat{a}$. We assume that the agent is in the exploitation phase and therefore plays the action with the highest expected reward. This feature importance is described by an importance-score or *saliency* for each feature $f$, denoted by $S$, where $S[f] \in (0, 1)$ denotes the saliency of the $f^{\text{th}}$ feature of $s$ for the agent taking action $\hat{a}$. A higher value indicates that the $f^{\text{th}}$ feature of $s$ is more important for the agent when taking action $\hat{a}$.

**Perturbation-based Saliency Maps**    The general outline of perturbation based saliency approaches is as follows. For each feature $f$, first perturb $s$ to get $s'$. For instance, in chess, we can perturb the board position by removing the piece in the $f^{\text{th}}$ square. In Atari, Greydanus et al. (2018) perturb the input image by adding a Gaussian blur centered on the $f^{\text{th}}$ pixel. Second, query $M$ to get $Q(s', a)$ $\forall a \in \mathcal{A}_s \cap \mathcal{A}_{s'}$. We take the intersection of $A_s$ and $A_{s'}$ to represent the case where some actions may be legal in $s$ but not in $s'$ and vice versa. For instance, when we remove a piece in chess, actions that were legal earlier may not be legal anymore. In the rest of this section, when we use "all actions" we mean all actions that are legal in both the states $s$ and $s'$. Finally, compute $S[f]$ based on how different $Q(s, a)$ and $Q(s', a)$) are, i.e. intuitively, $S[f]$ should be higher if $Q(s', a)$ is significantly different from $Q(s, a)$. Greydanus et al. (2018) compute the saliency map using $S_1[f] = \frac{1}{2}|\pi_s - \pi_{s'}|^2$, and $S_2[f] = \frac{1}{2}(V(s) - V(s'))^2$, while Iyer et al. (2018) use $S[f] = Q(s, \hat{a}) - Q(s', \hat{a})$. In this work, we will propose an alternative approach to compute $S[f]$.

**Properties**    We define two desired properties of an accurate saliency map for policy-based agents:

1. **Specificity:** Saliency $S[f]$ should focus on the effect of the perturbation *specifically* on the action being explained, $\hat{a}$, i.e. it should be high if perturbing the $f^{\text{th}}$ feature of the state reduces the relative expected reward of the selected action. Stated another way, $S[f]$ should be high if $Q(s, \hat{a}) - Q(s', \hat{a})$ is substantially higher than $Q(s, a) - Q(s', a), a \neq \hat{a}$. For instance, in figure 1, removing pieces such as the white queen impact all actions uniformly ($Q(s, a) - Q(s', a)$ is roughly equal for all actions). Therefore, such pieces should not be salient for explaining $\hat{a}$. On the other hand, removing pieces such as the white knight on a4 specifically impacts the move ($\hat{a}$ =Bb6) we are trying to explain ($Q(s, Bb6) - Q(s', Bb6) \gg Q(s, a) - Q(s', a)$ for other actions $a$). Therefore, such pieces should be salient for $\hat{a}$.

2. **Relevance:** Since the $Q$-values represent the expected returns, two states $s$ and $s'$ can have substantially different $Q$-values for all actions, i.e. may be higher for $s'$ for all actions if $s'$ is a *better* state. Saliency map for a specific action $\hat{a}$ in $s$ should thus ignore such differences, i.e. $s'$ should contribute to the saliency only if its effects are *relevant* to $\hat{a}$. In other words, $S[f]$ should be low if perturbing the $f^{\text{th}}$ feature of the state alters the expected rewards of actions other than $\hat{a}$. For instance, in Figure 1, removing the black pawn on c6 increases the expected reward of other actions (in this case, Bb4). However, it does not effect the expected reward of the action to be explained (Bb6). Therefore, the pawn is not salient for explaining the move (Bb6). In general, such features that are irrelevant to $\hat{a}$ should not be salient.

Existing approaches to saliency maps do not capture these properties in how they compute the saliency. Both the saliency approaches used in Greydanus et al. (2018), i.e. $S_1[f] = \frac{1}{2}(V(s) - V(s'))^2$ and $S_2[f] = \frac{1}{2}|\pi_s - \pi_{s'}|^2$, are not focusing on the action-specific effects since they aggregate the change over all actions. Although the saliency computation in Iyer et al. (2018) is somewhat more specific to the action, i.e. $S[f] = Q(s, \hat{a}) - Q(s', \hat{a})$, it is ignoring whether the effects on $Q$ are relevant only to $\hat{a}$, or effect all the other actions as well. This is illustrated in Figure 1.

**Identifying Specific Changes**    To focus on the effect of the change on the action, we are interested in whether the *relative* returns of $\hat{a}$ change with the perturbation. Using $Q(s, \hat{a})$ directly, as in Iyer et al. (2018), does not capture the relative changes to $Q(s, a)$ for other actions. To support specificity, we use the softmax over Q-values to normalize the values (as is also used in softmax action selection):

$$P(s, \hat{a}) = \frac{\exp(Q(s, \hat{a}))}{\sum_a \exp(Q(s, a))} \tag{1}$$

and compute $\Delta p = P(s, \hat{a}) - P(s', \hat{a})$, the difference in the relative expected reward of the action to be explained between the original and the perturbed state. A high value of $\Delta p$ thus implies that the feature is important for the *specific* choice of action $\hat{a}$ by the agent, while a low value indicates that the effect is not specific to the action.

**Identifying Relevant Changes**   Apart from focusing on the change in $Q(s, \hat{a})$, we also want to ensure that the perturbation leads to minimal effect on the relative expected returns for other actions. To capture this intuition, we will compute the relative returns of all other actions, and compute saliency in proportion to their similarity. Specifically, we normalize the Q-values using a softmax *apart* from the selected action $\hat{a}$.

$$P_{\text{rem}}(s, a) = \frac{\exp(Q(s, a))}{\sum_{a' \neq \hat{a}} \exp(Q(s, a'))} \quad \forall a \neq \hat{a} \tag{2}$$

We use the KL-Divergence $D_{KL} = P_{\text{rem}}(s', a) || P_{\text{rem}}(s, a)$ to measure the difference between $P_{\text{rem}}(s', a)$ and $P_{\text{rem}}(s, a)$. A high $D_{KL}$ indicates that the relative expected reward of taking some actions (other than the original action) changes significantly between $s$ and $s'$. In other words, a high $D_{KL}$ indicates that the effect of the feature is spread over other actions, i.e. the feature may not be *relevant* for the selected action $\hat{a}$.

**Computing the SARFA Saliency**   To compute salience $S[f]$, we need to combine $\Delta p$ and $D_{KL}$. If $D_{KL}$ is high, $S[f]$ should be low, regardless of whether $\Delta p$ is high; the perturbation is affecting many other actions. Conversely, when $D_{KL}$ is low, $S[f]$ should depend on $\Delta p$. To be able to compare these properties on a similar scale, we define a normalized measure of distribution *similarity* $K$ using $D_{KL}$:

$$K = \frac{1}{1 + D_{KL}} \tag{3}$$

As $D_{KL}$ goes from 0 to $\infty$, $K$ goes from 1 to 0. Thus, $S[f]$ should be low if either $\Delta p$ is low or $K$ is low. Harmonic mean provides this desired effect in a robust, smooth manner, and therefore we define $S[f]$ to be the harmonic mean of $\Delta p$ and $K$:

$$S[f] = \frac{2K\Delta p}{K + \Delta p} \tag{4}$$

Equation 4 captures our desired properties of saliency maps. If perturbing the $f^{\text{th}}$ feature affects the expected rewards of all actions uniformly, then $\Delta p$ is low and subsequently $S[f]$ is low. This low value of $\Delta p$ captures the property of *specificity* defined above. If perturbing the $f^{\text{th}}$ feature of the state affects the rewards of some actions other than the action to be explained, then $D_{KL}$ is high, $K$ is low, and $S[f]$ is low. This low value of $K$ captures the property of *relevance* defined above.

## 3   RESULTS

To show that SARFA produces more meaningful saliency maps than existing approaches, we use sample positions from Chess, Atari (Breakout, Pong and Space Invaders) and Go (Section 3.1). To show that SARFA generates saliency maps that provide useful information to humans, we conduct human studies on problem-solving for chess puzzles (Section 3.2). To automatically compare the saliency maps generated by different perturbation-based approaches, we introduce a Chess saliency dataset (Section 3.3). We use the dataset to show how SARFA is better than existing approaches in identifying chess pieces that humans deem relevant in several positions. In Section 3.4, we show how SARFA can be used to understand common tactical ideas in chess by interpreting the action of a trained agent.

To show that SARFA works for black-box agents, regardless of whether they are trained using reinforcement learning, we use a variety of agents. We only assume access to the agent's $Q(s, a)$ function for all experiments. For experiments on chess, we use the Stockfish agent[3]. For experiments on Go, we use the pre-trained MiniGo RL agent[4]. For experiments on Atari agents and for generating saliency maps for Greydanus et al. (2018), we use their code and pre-trained RL agents[5]. For generating saliency maps using Iyer et al. (2018), we use our own implementation[6]. All of our code and more detailed results are available in our Github repository: https://nikaashpuri.github.io/sarfa-saliency/.

---

[3] https://stockfishchess.org/
[4] https://github.com/tensorflow/minigo
[5] https://github.com/greydanus/visualize_atari

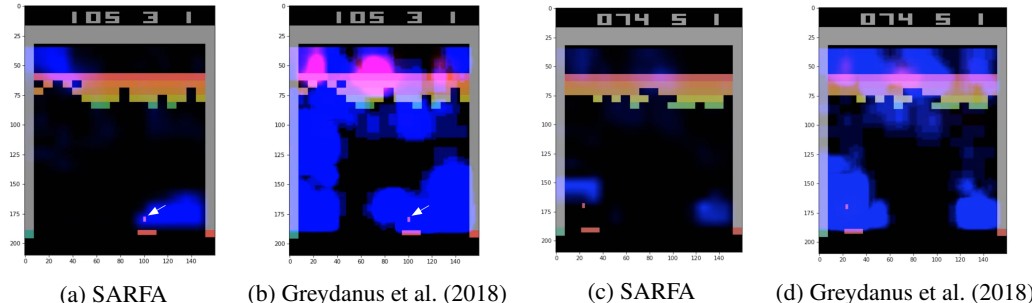

(a) SARFA            (b) Greydanus et al. (2018)            (c) SARFA            (d) Greydanus et al. (2018)

Figure 2: Comparing saliency of RL agents trained to play Breakout

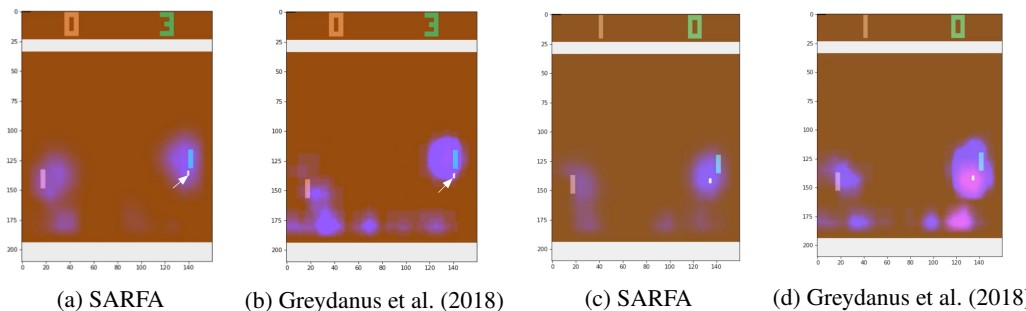

(a) SARFA            (b) Greydanus et al. (2018)            (c) SARFA            (d) Greydanus et al. (2018)

Figure 3: Comparing saliency of RL agents trained to play Atari Pong

## 3.1 ILLUSTRATIVE EXAMPLES

In this section, we provide examples of generated saliency maps to highlight the qualitative differences between SARFA that is action-focused and existing approaches that are not.

**Chess** Figure 1 shows sample positions where SARFA produces more meaningful saliency maps than existing approaches for a chess-playing agent (Stockfish). Greydanus et al. (2018) and Iyer et al. (2018) generate saliency maps that highlight pieces that are not relevant to the move played by the agent. This is because they use differences in $Q(s, a)$, $V(s)$ or the the $L_2$ norm of the policy vector between the original and perturbed state to calculate the saliency maps. Therefore, pieces such as the white queen that affect the value estimate of the state are marked salient. In contrast, the saliency map generated by SARFA only highlights pieces relevant to the move.

**Atari** To show that SARFA generates saliency maps that are more focused than those generated by Greydanus et al. (2018), we compare the approaches on three Atari games: Breakout, Pong, and Space Invaders. Figures 2, 3, and 4 shows the results. SARFA highlights regions of the input image more precisely, while the Greydanus et al. (2018) approach highlights several regions of the input image that are not relevant to explain the action taken by the agent.

**Go** Figure 5 shows a board position in Go. It is black's turn. The four white stones threaten the three black stones that are in one row at the top left corner of the board. To save those three black stones, black looks at the three white stones that are directly below the three black ones. Due to another white stone below the three white stones, the continuous row of three white stones cannot be captured easily. Therefore black moves to place a black stone below that single white stone in an attempt to start capturing the four white stones. It takes the next few turns to surround the structure of four white stones with black ones, thereby saving its pieces. The method described in Greydanus et al. (2018) generates a saliency map that highlights almost all the pieces on the board. Therefore, it reveals little about the pieces that the agent thinks are important. On the other hand, the map produced by Iyer et al. (2018) highlights only a few pieces. The saliency map generated by SARFA correctly highlights the structure of four white stones and the black stones already present around them that may be involved in capturing them.

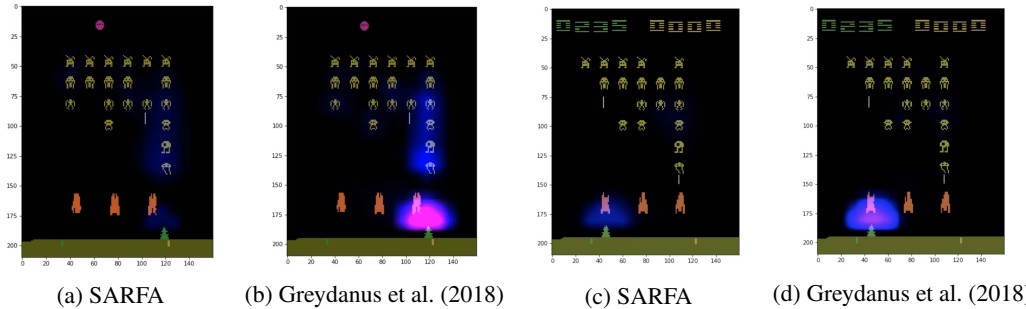

(a) SARFA      (b) Greydanus et al. (2018)      (c) SARFA      (d) Greydanus et al. (2018)

Figure 4: Comparing saliency of RL agents trained to play Space Invaders

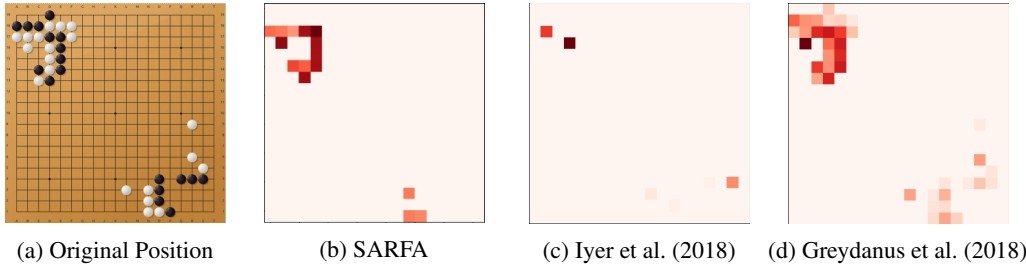

(a) Original Position      (b) SARFA      (c) Iyer et al. (2018)      (d) Greydanus et al. (2018)

Figure 5: Comparing saliency maps generated by different approaches for the MiniGo agent

## 3.2 HUMAN STUDIES: CHESS

To show that SARFA generates saliency maps that provide useful information to humans, we conduct human studies on problem-solving for chess puzzles. We show forty chess players (ELO 1600-2000) fifteen chess puzzles from https://www.chess.com (average difficulty ELO 1800). For each puzzle, we show either the puzzle without a saliency map, or the puzzle with a saliency map generated by SARFA, Greydanus et al. (2018), or Iyer et al. (2018). The player is then asked to solve the puzzle. We measure the accuracy (number of puzzles correctly solved) and the average time taken to solve the puzzle, shown in Table 1. The saliency maps generated by SARFA are more helpful for humans when solving puzzles than those generated by other approaches. We observed that the saliency maps generated by Greydanus et al. (2018) often confuse humans, because they highlight several pieces unrelated to the tactic. The maps generated by Iyer et al. (2018) highlight few pieces and therefore are marginally better than showing no saliency maps for solving puzzles.

## 3.3 CHESS SALIENCY DATASET

To automatically compare the saliency maps generated by different perturbation-based approaches, we introduce a Chess saliency dataset. The dataset consists of 100 chess puzzles[7]. Each puzzle has a single correct move. For each puzzle, we ask three human experts (ELO > 2200) to mark the pieces that are important for playing the correct move. We take a majority vote of the three experts to obtain a list of pieces that are important for the move played in the position. The complete dataset is available in our Github repository[6]. We use this dataset to compare SARFA to existing approaches

---

[6]https://nikaashpuri.github.io/sarfa-saliency/

Table 1: Results of Human Studies for solving chess puzzles

|  | No Saliency | SARFA | Greydanus et al. | Iyer et al. |
|---|---|---|---|---|
| **Accuracy** | 56.67% | **72.41%** | 40.84% | 24.60% |
| **Average time taken** | 77.53 sec | **67.02 sec** | 70.95 sec | 102.26 sec |

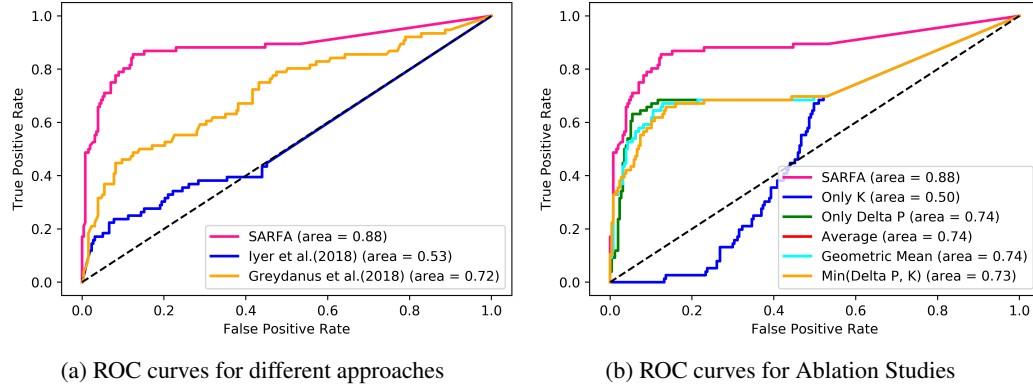

(a) ROC curves for different approaches          (b) ROC curves for Ablation Studies

Figure 6: ROC curves comparing approaches on the chess saliency dataset

(Greydanus et al., 2018; Iyer et al., 2018). Each approach generates a list of squares and a score that indicates how salient the piece on the square is for a particular move. We scale the scores between 0 and 1 to generate ROC curves. Figure 6a shows the results. SARFA generates saliency maps that are better than existing approaches at identifying chess pieces that humans deem relevant in certain positions.

To evaluate the relative importance of the two components in our saliency computation ($S[f]$; Equation 4), we compute saliency maps and ROC curves using each component individually, i.e. $S[f] = \Delta p$ or $S[f] = K$, and compare harmonic mean to other ways to combine them, i.e. using the average, geometric mean, and minimum of $\Delta p$ and $K$. Figure 6b shows the results. Combination of the two properties via harmonic mean leads to more accurate saliency maps than alternative approaches.

### 3.4 EXPLAINING TACTICAL MOTIFS IN CHESS

In this section, we show how SARFA can be used to understand common tactical ideas in chess by interpreting the action of a trained agent. Figure 7 illustrates common tactical positions in chess. The corresponding saliency maps are generated by interpreting the moves played by the Stockfish agent in these positions.

In Figure 7a, it is white to move. The surprising Rook x d6 is the move played by Stockfish. Figure 7d shows the saliency map generated by SARFA. The map illustrates the key idea in the position. Once black's rook recaptures white's rook, white's bishop pins it to the black king. Therefore, white can increase the number of attackers on the rook. The additional attacker is the pawn on e4 highlighted by the saliency map.

In Figure 7b, it is white to move. Stockfish plays Queen x h7. A queen sacrifice! Figure 7e shows the saliency map. The map highlights the white rook and bishop, along with the queen. The key idea is that once black captures the queen with his king (a forced move), then the white rook moves to h5 with checkmate. This checkmate is possible because the white bishop guards the important escape square on g6. The saliency map highlights both pieces.

In Figure 7c, it is black to move. Stockfish plays the sacrifice rook x d4. The saliency map in Figure 7f illustrates several key aspects of the position. The black queen and light-colored bishop are threatening mate on g2. The white queen protects g2. The white rook on a5 is unguarded. Therefore, once white recaptures the sacrificed rook with the pawn on c3, black can attack both the white rook and queen with the move bishop to b4. The idea is that the white queen is "overworked" or "overloaded" on d2, having to guard both the g2-pawn and the a5-Rook against black's double attack.

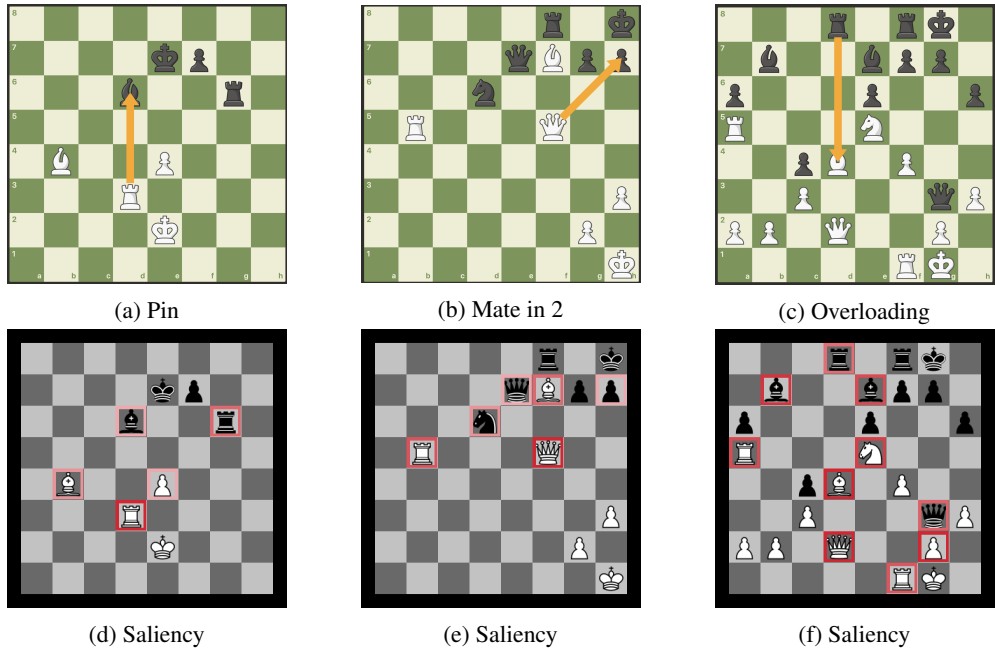

|                  |                  |                  |
| :--------------: | :--------------: | :--------------: |
| (a) Pin          | (b) Mate in 2    | (c) Overloading  |
| (d) Saliency     | (e) Saliency     | (f) Saliency     |

Figure 7: Saliency maps generated by SARFA that demonstrate common tactical motifs in chess

## 3.5    Robustness to Perturbations

We are also interested in evaluating the robustness of the generated saliency maps: is the saliency different if non-salient changes are made to the state? To evaluate the robustness of SARFA, we perform two irrelevant perturbations to the positions in the chess saliency dataset. First, we pick a random piece amongst the ones labeled non-salient by *human experts* in a particular position, and remove it from the board. We repeat this for each puzzle in the dataset to generate a new perturbed saliency dataset. Second, we remove a random piece amongst ones labeled non-salient by *SARFA* for each puzzle, creating another perturbed saliency dataset. In order to evaluate the effect of non-salient perturbations on our generated saliency maps, we compute the AUC values for the generated saliency maps, as above, for these perturbed datasets. Since we remove non-salient pieces, we expect the saliency maps and subsequently AUC value to be similar to the value on the original dataset. For both these perturbations, we get an AUC value of *0.92*, same as the value on the non-perturbed dataset, confirming the robustness of our saliency maps to these non-relevant perturbations.

## 4    Related Work

Since understanding RL agents is important both for deploying RL agents to the real world and for gaining insights about the tasks, a number of different kinds of interpretations have been introduced.

A number of approaches generate natural language explanations to explain RL agents (Dodson et al., 2011; Elizalde et al., 2008; Khan et al., 2009). They assume access to an exact MDP model and that the policies map from interpretable, high-level state features to actions. More recently, Hayes & Shah (2017) analyze execution traces of an agent to extract explanations. A shortcoming of this approach is that it explains policies in terms of hand-crafted state representations that are semantically meaningful to humans. This is often not practical for board games or Atari games where the agents learn from raw board/visual input. Zahavy et al. (2016) apply t-SNE (Maaten & Hinton, 2008) on the last layer of a deep Q-network (DQN) to cluster states of behavior of the agent. They use Semi-Aggregated Markov Decision Processes (SAMDPs) to approximate the black box RL policies. They use the more interpretable SAMDPs to gain insight into the agent's policy. An issue with the explanations is that they emphasize t-SNE clusters that are difficult to understand for non-experts. To build user trust and increase adoption, it is important that the insight into agent behavior should be in a form that is interpretable to the untrained eye and obtained from the original policy instead of a distilled one.

Most relevant to SARFA are the visual interpretable explanations of deep networks using saliency maps. Methods for computing saliency can be classified broadly into two categories.

*Gradient-based methods* identify input features that are most salient to the trained DNN by using the gradient to estimate their influence on the output. Simonyan et al. (2013) use gradient magnitude heatmaps, which was expanded upon by more sophisticated methods to address their shortcoming, such as guided backpropagation (Springenberg et al., 2014), excitation backpropagation (Zhang et al., 2018), DeepLIFT (Shrikumar et al., 2017), GradCAM (Selvaraju et al., 2017), and GradCAM++ (Chattopadhay et al., 2018). Integrate gradients (Sundararajan et al., 2017) provide two *axioms* to further define the shortcomings of these approaches: sensitivity (relative to a baseline) and implementation invariance, and use them to derive an approach. Nonetheless, all gradient-based approaches still depend on the shape in the immediate neighborhood of a few points, and conceptually, use perturbations that lack physical meaning, making them difficult to use and vulnerable to adversarial attacks in form of imperceivable noise (Ghorbani et al., 2019). Further, they are not applicable to scenarios with black-box access to the agent, and even with white-box access to model internals, they are not applicable when agents are not fully differentiable, such as Stockfish for chess.

We are more interested in *perturbation-based* methods for black-box agents: methods that compute the importance of an input feature by removing, altering, or masking the feature in a domain-aware manner and observing the change in output. It is important to choose a perturbation that removes information without introducing any new information. As a simple example, Fong & Vedaldi (2017) consider a classifier that predicts 'True' if a certain input image contains a bird and 'False' otherwise. Removing information from the part of the image which contains the bird should change the classifier's prediction, whereas removing information from other areas should not. Several kinds of perturbations have been explored, e.g. Zeiler & Fergus (2014); Ribeiro et al. (2016) remove information by replacing a part of the input with a gray square. Note that these approaches are implementation invariant by definition, and are sensitive with respect to the perturbation function.

Existing perturbation-based approaches for RL (Greydanus et al., 2018; Iyer et al., 2018), however, by focusing on the complete $Q$ (or $V$), tend to produce saliency maps that are not specific to the action of interest. SARFA addresses this by measuring the impact *only* on the action being selected, resulting in more focused and useful saliency maps, as we show in our experiments.

## 5 Limitations and Future Work

Saliency maps focus on visualizing the dependence between the input and output to the model, essentially identifying the *situation-specific* explanation for the decision. Although such *local* explanations have applications in understanding, debugging, and developing trust with machine learning systems, they do not provide any direct insights regarding the general behavior of the model, or guarantee that the explanation is applicable to a different scenario. Attempts to provide a more general understanding of the model include carefully selecting prototype explanations to show to the user (van der Linden et al., 2019) and crafting explanations that are precise and actionable (Ribeiro et al., 2018). We will explore such ideas for the RL setting in future, to provide explanations that accurately characterize the behavior of the policy function, in a precise, testable, and intuive manner.

There are a number of limitations of SARFA to generating saliency maps in our current implementation. First, we perturb the state by removing information (removing pieces in Chess/Go, blurring pixels in Atari). Therefore, SARFA cannot highlight the importance of *absence* of certain attributes, i.e. saliency of certain positions being empty. In games such as Chess and Go, an empty square or file (collection of empty squares) can often be important for a particular move. Future work will explore perturbation functions that add information to the state (e.g. adding pieces in Chess/Go). Such functions, along with SARFA, can be used to calculate the importance of empty squares. Second, it is possible that perturbations may explore states that lie outside the manifold, i.e. they lead to invalid states. For example, unless explicitly prohibited like we do, SARFA will compute the saliency of the king pieces by removing them, which is not allowed in the game, or remove the paddle from Pong. In future, we will explore strategies that take the valid state space into account when computing the saliency. Last we estimate the saliency of each feature independently, ignoring feature dependencies and correlations, which may lead to incorrect saliency maps. We will investigate approaches that perturb multiple features to quantify the importance of each feature (Ribeiro et al., 2016; Lundberg & Lee, 2017), and combine them with SARFA to explaining black-box policy-based agents.

## 6 Conclusion

We presented a perturbation-based approach that generates more focused saliency maps than existing approaches by balancing two aspects (specificity and relevance) that capture different desired characteristics of saliency. We showed through illustrative examples (Chess, Atari, Go), human studies (Chess), and automated evaluation methods (Chess) that SARFA generates saliency maps that are more interpretable for humans than existing approaches. The results of our technique show that saliency can provide meaningful insights into a black-box RL agent's behavior. For the code release and demo videos, see https://nikaashpuri.github.io/sarfa-saliency/.

## Acknowledgements

We would like to thank the anonymous reviewers for their helpful comments and suggestions. This work is supported in part by the NSF Award No. IIS-1756023 and in part by a gift from the Allen Institute of Artificial Intelligence (AI2).

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

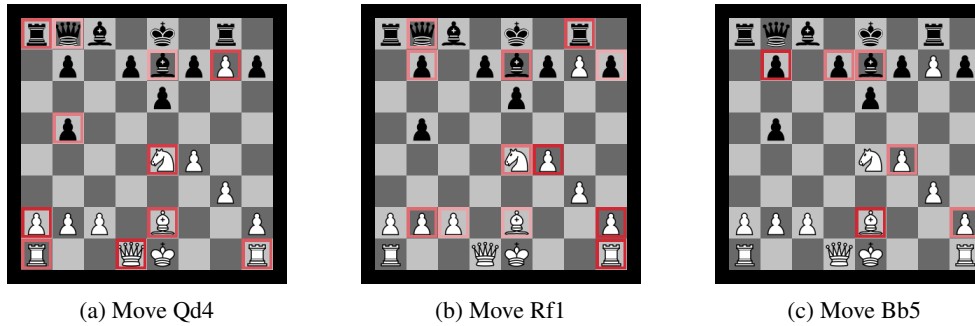

(a) Move Qd4              (b) Move Rf1              (c) Move Bb5

Figure 8: Saliency Maps generated by SARFA for the top 3 moves in a chess position

## A  EXPERIMENTAL DETAILS

For experiments on chess, we use the Stockfish 10 agent: https://stockfishchess.org/. Stockfish works using a heuristic-based measure for each state along with Alpha-Beta Pruning to search over the state-space.

For experiments on Go, we use the pre-trained MiniGo RL agent: https://github.com/tensorflow/minigo. This agent was trained using the AlphaGo Algorithm (Silver et al., 2016). It also adds features and architecture changes from the AlphaZero Algorithm Silver et al. (2017).

For experiments on Atari agents and for generating saliency maps for Greydanus et al. (2018), we use their code and pre-trained RL agents available at https://github.com/greydanus/visualize_atari. These agents are trained using the Asynchronous Advantage Actor-Critic Algorithm (A3C) (Mnih et al., 2016).

For generating saliency maps using Iyer et al. (2018), we use our implementation. All of our code and more detailed results are available in our Github repository: https://nikaashpuri.github.io/sarfa-saliency/ .

For chess and Go, we perturb the board position by removing one piece at a time. We do not remove a piece if the resulting position is illegal. For instance, in chess, we do not remove the king. For Atari, we use the perturbation technique described in Greydanus et al. (2018). The technique perturbs the input image by adding a Gaussian blur localized around a pixel. The blur is constructed using the Hadamard product to interpolate between the original input image and a Gaussian blur. The saliency maps for Atari agents have been computed on the frames provided by Greydanus et al. (2018) in their code repository.

The puzzles for conducting the Chess human studies, creating the Chess Saliency Dataset, and providing illustrative examples have been taken from Lichess: https://database.lichess.org/. The puzzles for illustrative examples on Go have been taken from OnlineGo: https://online-go.com/puzzles.

## B  SALIENCY MAPS FOR TOP 3 MOVES

Figure 8 shows the saliency maps generated by SARFA for the top 3 moves in a chess position. The maps highlight the different pieces that are salient for each move. For instance, Figure 8a shows that for the move Qd4, the pawn on g7 is important. This is because the queen move protects the pawn. For the saliency maps in Figures 8b and 8c, the pawn on g7 is not highlighted.

## C  SALIENCY MAPS FOR LEELAZERO

To show that SARFA generates meaningful saliency maps in Chess for RL agents, we interpret the LeelaZero Deep RL agent https://github.com/leela-zero/leela-zero. Figure 9 shows the results. As discussed in Section 1, the saliency maps generated by (Greydanus et al., 2018; Iyer et al., 2018)

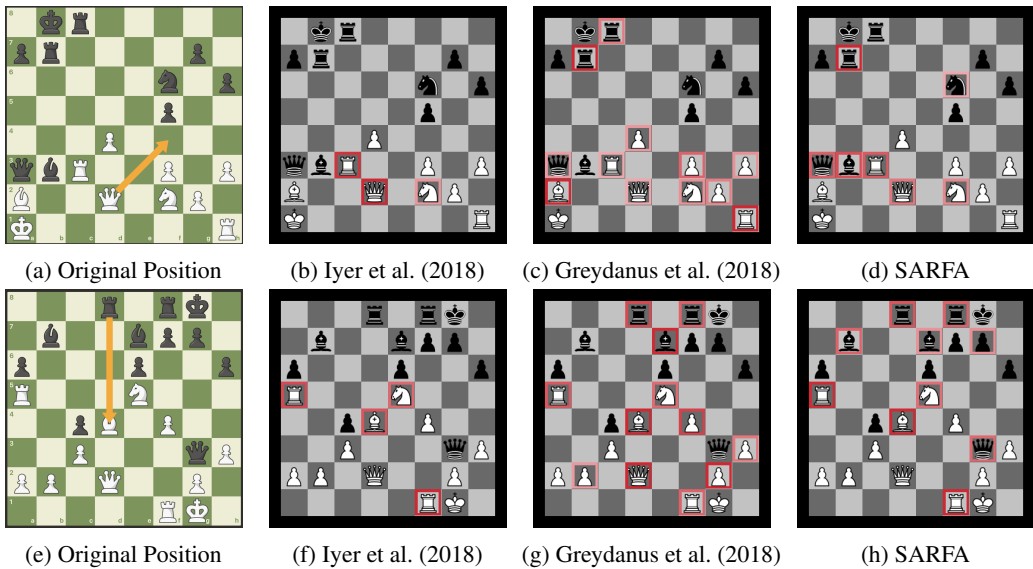

Figure 9: Saliency maps generated by different approaches for the LeelaZero Deep Reinforcement Learning Agent

highlight several pieces that are not relevant to the move being explained. On the other hand, the saliency maps generated by SARFA highlight the pieces relevant to the move.

