# OpenReview forum: "Explain Your Move: Understanding Agent Actions Using Specific and Relevant Feature Attribution"
_ICLR.cc/2020/Conference — Accept (Poster)_

### Official Review · AnonReviewer1 · 2019-10-05
**Official Blind Review #1**

**Rating:** 8

**Review:**

Interpreting the policies of RL agents is an important consideration if we would like to actually deploy them and understand their behaviours. Prior works have applied saliency methods to DRL agents, but the authors note that there are two properties - specificity and relevance - that these methods do not take into account, and therefore result in misleading saliency maps. The authors define (and provide examples) of these properties, propose simple ways to calculate these (and like prior methods, relying on perturbations and therefore applicable to almost black box agents), and combine them neatly using the harmonic mean to provide a new way to calculate saliency maps for agents with discrete action spaces. While the improvements on Atari are hard to quantify, the results on chess and go are more interpretable and hence more convincing. The study using saliency maps to aid human chess players is rather neat, and again adds evidence towards the usefulness of this technique. Finally, the chess saliency dataset is an exciting contribution that can actually be used to quantify saliency methods for DRL agents. The proposed method is relatively simple, but is well-motivated and demonstratively (quantitatively, which is great work for general interpretability methods) better than prior methods, and in addition the authors introduce a new dataset + quantitative measure for saliency methods, so I would give this paper an accept.

Although the authors motivate their choice of perturbation-based saliency methods as opposed to gradient-based methods, they should expand their review of the latter. As the technique the authors introduced is very general, it would be useful to know how it compares to the current state of research in terms of identifying properties that attribution methods should meet - a good example of this is integrated gradients (Sundararajan et al., 2017), which similarly identify "sensitivity" and "implementation invariance" as "axioms" that their method satisfies.

**Experience Assessment:**

I have read many papers in this area.

**Review Assessment: Checking Correctness Of Derivations And Theory:**

N/A

**Review Assessment: Checking Correctness Of Experiments:**

I carefully checked the experiments.

**Review Assessment: Thoroughness In Paper Reading:**

I read the paper thoroughly.

---

> ### Author Response · Authors · 2019-11-09
> **Comparison to gradient-based approaches**
>
> We have expanded the review of these approaches in the revised Section 4, including the connection of the axioms of attribution to perturbation-based techniques. More empirically speaking, Greydanus et al compared their approach against a gradient-based approach, and they noted in their paper that such saliency maps “can be difficult to interpret“, “choose perturbations which lack physical meaning”, and “may move it off from the manifold of realistic input images”. In our paper, we show that our explanations are more useful than Greydanus et al. Note that the gradient-based techniques not only need white-box access to the agent, but even with white-box access, some agents may not be fully differentiable, such as Stockfish for chess.

---

> > ### Comment · AnonReviewer1 · 2019-11-12
> > **Perturbations**
> >
> > Thank you for adding this line of related work. Your criticisms are perfectly valid. Though, for what it's worth, having working out a "meaningful" perturbation may be difficult depending on the domain, so this is both a pro and a con.

---

> > > ### Author Response · Authors · 2019-11-15
> > > **Perturbation and Gradient-Based methods**
> > >
> > > Agreed, both perturbation and gradient methods are useful for different use cases; we mention the difficulty in identifying meaningful perturbations in Section 5.

---

### Official Review · AnonReviewer3 · 2019-10-22
**Official Blind Review #3**

**Rating:** 8

**Review:**

This paper proposes an algorithm for explaining the move of the agents trained by reinforcement learning (RL) by generating a saliency map.
The authors proposed two desired properties for the saliency map, specificity and relevance.
The authors then pointed out that prior studies failed to capture one of the two properties.
To combine the two components into a single saliency map, the authors proposed using the harmonic mean.

The experimental results demonstrated that the proposed saliency map successfully focused only on important parts while the other method tend to highlight some irrelevant parts also.
The authors also did a great job for evaluating the goodness of the saliency maps, by preparing a human annotated chess puzzle dataset.

I think the paper is well-written, and the basic idea look reasonable and promising.
The experimental evaluations are well designed and the results look convincing.
Saliency map for RL is not yet mature, and I expect to see further improvements (especially, more theoretically principled ones) follow this study.


### Updated after author response ###
The response from the authors seem to be reasonable to me. I therefore keep my score.

**Experience Assessment:**

I do not know much about this area.

**Review Assessment: Checking Correctness Of Derivations And Theory:**

I assessed the sensibility of the derivations and theory.

**Review Assessment: Checking Correctness Of Experiments:**

I assessed the sensibility of the experiments.

**Review Assessment: Thoroughness In Paper Reading:**

I made a quick assessment of this paper.

---

> ### Author Response · Authors · 2019-11-09
> **Thank you for the positive feedback**
>
> We would like to thank the reviewer for the positive feedback. We will work on further improving the quality of the paper over subsequent versions.

---

> > ### Comment · AnonReviewer3 · 2019-11-13
> > **A small question**
> >
> > I would like to thank the authors for the reply.
> > I have a small question if there is any desired properties other than Specificity and Relevance.
> > Some studies (e.g. SHAP, IntegratedGrad) considered these desired properties as "axioms", and developed saliency methods as the ones that satisfy the axioms.
> > I am interested in to see if a similar axiomatic approach is applicable for this study.

---

> > > ### Author Response · Authors · 2019-11-15
> > > **Additional desired properties**
> > >
> > > Thank you for your comment. Based on our initial evaluation, it is very challenging to derive a scalable approach purely based on using specificity and relevance as the “axioms”, due to the need to evaluate exponentially many subsets of features, made much worse due to the fact that many such perturbations will be invalid or not meaningful. However, this direction is definitely useful, and we will explore this in future work.
> > >
> > > In regards to whether specificity and relevance are sufficient, and whether there are additional desired properties for saliency, our extensive experiments with humans for puzzle-solving and on the chess saliency dataset show that our approach (specificity + relevance) performs significantly better than existing approaches. There may be other additional properties though, especially in light of the discussion of the limitations in Section 5, that we will explore in future work.

---

### Official Review · AnonReviewer2 · 2019-10-22
**Official Blind Review #2**

**Rating:** 6

**Review:**

[score raised from weak reject to weak accept after rebuttal - on a more fine-grained scale, I would rate this paper now an accept (7), but not a strong accept (8), however since this year's scale is quite coarse, I'll stick with a score of 6]

Summary
The paper proposes a new perturbation-based measure for computing input-saliency maps for deep RL agents. As reported in a large body of literature before, such saliency maps are supposed to help “explain” why a deep RL system picked a certain action. The measure proposed in the paper aims at combining two aspects: specificity and relevance, which should ensure that the saliency map highlights inputs that are relevant for a particular action to be explained (specificity), and this particular action only (relevance). The paper shows illustrative examples of the proposed approach and two previously proposed alternatives on Chess, Go and three Atari games. Additionally the paper evaluates the method and the two previous alternatives on two interesting chess-tasks: chess-puzzles where human players were shown to be able to solve puzzles faster and with higher accuracy when given the proposed saliency map in addition, and evaluation against a curated dataset of human-expert saliency maps for 100 chess puzzles.

Contributions
i) Novel, perturbation-based saliency measure composed of two parts. Main idea of specificity is (similar to Iyer et al. 2018) to use State-Action value function (Q-function) with a specific action, instead of State-Value function only. Main idea of relevance is to “normalize” by taking into account change in Q-value for all other actions as well. Both parts are combined in a heuristic fashion.

ii) More objective/reproducible assessment of how saliency maps produced by different methods overlap with human judgement of saliency of pieces in chess. To this end: two experiments with human chess players/experts (puzzle solving, and expert-designed saliency maps).

Quality, Clarity, Novelty, Impact
The paper is well written and most sections are easily understandable, though for some parts it might help if the reader is quite familiar with Chess/Go. The proposed saliency measure seems to address some shortcomings of previously proposed measures - my main criticism is that the construction of the measure seems rather ad-hoc and heuristic. It would have been great to formally define specificity and relevance (e.g. in an information-theoretic framework as Sparse Relevant Information) and then derive a suitable measure that is shown to satisfy/approximate the formal definitions. At least, there is one ablation study that justifies parts of the heuristic construction to some degree.
The proposed measure seems to do reasonably well on board-game domains, in particular chess. However it might also be the case that the measure does particularly well for generating saliency maps for Stockfish (which is the agent that happens to be evaluated in the chess domain), which might be quite different from previously reported methods that have been designed for deep neural network RL agents. The illustrative examples on Atari and Go do not allow for a statistically significant judgement of the quality of the proposed method.

On a conceptual level, a bigger issue (of many saliency methods in general, but the criticism also applies to the paper) is that the “explanations” drawn from saliency maps are rarely evaluated rigorously. The paper makes a nice attempt by trying to establish some “ground-truth” saliency in chess using humans to increase the degree of objectivity, which I greatly appreciate. However, it remains unclear whether explanations that happen to coincide with human notions of saliency really are of higher quality for assessing how an artificial system makes its decisions. The main goal of explainability/interpretability methods must be to come up with testable hypotheses that tell us something about how the artificial system makes its decisions in novel/unseen situations. The goal is not to explain a decision mechanistically after the fact (which is trivial, given a deterministic, differentiable feed-forward system), but to come up with non-trivial explanations that extrapolate/generalize. Specificity and Relevance are probably important ingredients of such explanations, but I think it’s important to formalize them properly first. Currently I am in favor of suggesting a major revision of the work, but I am happy to reconsider my decision based on the rebuttal and other reviewers’ assessment. Having said that, I do want to reiterate that I think it is great that the authors included some ablation studies and measures of “usefulness” of the saliency method.

Improvements / Major comments
i) Formally define specificity and relevance (e.g. as sparsity and compression?). Ideally derive a saliency measure based on these formal definitions.

ii) Show how saliency maps (of the same situation) change when producing explanations for different actions. I assume that currently the illustrative examples only show the action with the highest Q-value, what changes when using e.g. a less likely action?

iii) Show that the saliency map stays roughly constant for seemingly irrelevant perturbations. In particular, using the chess-dataset with expert annotations, apply various kinds of perturbations to non-salient pieces (e.g. removing them, swapping them for other pieces and potentially moving them in irrelevant ways) and see whether the AUC stays roughly constant.

iv) Apply non-relevant perturbations to the salient pieces. I.e. take the same puzzle/scene and move it across the board, does the saliency-map move in (roughly) the same way.

v) The saliency method might be particularly suited for Stockfish (whose action-value estimates might be strongly influenced by human saliency and chess theory). See whether the method still produces good results for other chess agents (ideally trained without human heuristics or data). If this is hard to do for chess, think about a different application where this is easier.

vi) Add a section that discusses current shortcomings and caveats with the method, and saliency maps for explainability in general.

vii) (Experimental details) For each domain, please explain the perturbations that you used (removing pieces in Chess/Go, blurring pixels in Go, anything else e.g. replacing pieces?). In all experiments, was it always the action with the highest Q-value that was being explained?



Minor comments

a) Table 1: (add error-bars) What is the variance across players? Are the results for the proposed method statistically significant?

b) Chess saliency dataset. Are the expert saliency ratings binary? Why not have multiple degrees of saliency?

c) Would the Greydanus et al. 2018 approach deliver similar results when using a threshold to cut off low-saliency inputs?

d) Why is the saliency map in 3.4 binary (pieces are either salient or not)? How was the binarization threshold chosen? What would the non-binary saliency maps look like?

e) Please provide all experimental details (in the appendix) that are necessary to reproduce the experiments. Referring to a code-repository is not a replacement for describing the methods in detail.

**Experience Assessment:**

I have read many papers in this area.

**Review Assessment: Checking Correctness Of Derivations And Theory:**

I assessed the sensibility of the derivations and theory.

**Review Assessment: Checking Correctness Of Experiments:**

I assessed the sensibility of the experiments.

**Review Assessment: Thoroughness In Paper Reading:**

I read the paper at least twice and used my best judgement in assessing the paper.

---

> ### Author Response · Authors · 2019-11-09
> **Response to the comments**
>
> We would like to thank the reviewer for their comments, many of the comments are very useful and will address them in the following revisions.
>
> > saliency maps are rarely evaluated rigorously.
>
> Indeed, evaluating saliency maps is a very challenging problem. Thank you for appreciating our attempt at evaluation; note that it is fairly comprehensive compared to most of the published approaches. We included the lack of generalization in Section 5.
>
> > The main goal of explainability/interpretability methods must be to come up with testable hypotheses that tell us something about how the artificial system makes its decisions in novel/unseen situations.
>
> One of the goals of interpretability is indeed to describe the extrapolation/generalization of the agent, however, the goals of interpretability are multifold. Just knowing the reasons behind a single prediction/action can be quite useful, as we show. Note that this is aligned with the existing literature in “local explanations”, that has been shown to be useful for interpreting, debugging, developing trust, and for the users learning the task. However, we agree with the limitations, and include it in Section 5.
>
> > iv) Apply non-relevant perturbations to the salient pieces. I.e. take the same puzzle/scene and move it across the board, does the saliency-map move in (roughly) the same way.
>
> Unfortunately, it is not easy to identify non-relevant perturbations, since moving the relevant pieces across the board, for example, may completely change the underlying strategy that is applicable, making different pieces salient or not salient.
>
> > vi) Add a section that discusses current shortcomings and caveats with the method, and saliency maps for explainability in general.
>
> We have added such a section (Section 5).
>
> > vii) (Experimental details) For each domain, please explain the perturbations that you used (removing pieces in Chess/Go, blurring pixels in Go, anything else e.g. replacing pieces?).
> > e) Please provide all experimental details (in the appendix) that are necessary to reproduce the experiments. Referring to a code-repository is not a replacement for describing the methods in detail.
>
> We have added such a section (Appendix A).
>
> > In all experiments, was it always the action with the highest Q-value that was being explained?
>
> Yes, we only explain the actions with the highest Q-value, since greedy policies are often used at test time. However, we have also included examples of saliency maps for other actions, in Appendix B.
>
> > b) Chess saliency dataset. Are the expert saliency ratings binary? Why not have multiple degrees of saliency?
>
> Yes, the saliency ratings are binary. Different degrees of saliency is harder to obtain. For instance, in a Chess position, it is much easier for experts to label whether pieces are important than to estimate (or agree on) the degree of importance.
>
> > c) Would the Greydanus et al. 2018 approach deliver similar results when using a threshold to cut off low-saliency inputs?
>
> We feel that the “thresholding” should be part of the explanation technique; they should provide a sparse representation of which features are relevant (all of the baselines do already zero out irrelevant features). It is difficult to evaluate different thresholding amongst the features that they have identified as salient, since the threshold itself may be crucially dependent on the state or the value function, and it is not possible to search the different threshold values at each state. Nonetheless, we do provide an evaluation that is independent of the effect of the threshold (using ROC curves) but instead depends on the ranking of the features.
>
> > d) Why is the saliency map in 3.4 binary (pieces are either salient or not)? How was the binarization threshold chosen? What would the non-binary saliency maps look like?
>
> We have updated the saliency maps in Section 3.4 to the original non-binary saliency maps output by our approach.

---

> > ### Comment · AnonReviewer2 · 2019-11-14
> > **Thank you for the thorough response and the improvements to the manuscript**
> >
> > Most of my issues have been addressed to a satisfying degree during rebuttal. I will therefore raise my score accordingly. As I stated before, the main strength of the paper to me is the effort that the authors made to evaluate the quality of the produced explanations. Evaluations of this kind (and rigor) are severely lacking from large bodies of work on interpretability/saliency.
> >
> > I would still like to encourage the authors to try and come up with some perturbations that should have very predictable effects on the saliency (such as adding non-relevant pieces in non-relevant positions, or moving all pieces on the board by two rows/columns, or "mirroring" a scenario along the vertical axis as well as swapping colors). While this is tricky to do in full generality, I am sure that it is possible for particular situations. In the best case it would make the results even stronger, otherwise it would add some critical reflection and current shortcomings to be addressed in future work.

---

> > > ### Author Response · Authors · 2019-11-15
> > > **Additional changes**
> > >
> > > Thank you for your comments. We have made some additional changes in the latest revision.
> > >
> > > > the measure does particularly well for generating saliency maps for Stockfish
> > > > might be quite different … for deep neural network RL agents.
> > > > v) The saliency method might be particularly suited for Stockfish (whose action-value estimates might be strongly influenced by human saliency and chess theory).
> > > > See whether the method still produces good results for other chess agents
> > >
> > > Added saliency maps for a deep learning agent for Chess to Appendix C.
> > >
> > > > iii) Show that the saliency map stays roughly constant for seemingly irrelevant perturbations.
> > > > I would still like to encourage the authors to try and come up with some perturbations that should have very predictable effects on the saliency
> > >
> > > Added results in Section 3.5

---

### Author Response · Authors · 2019-11-09
**Revised version with additional content**

Thank you for the helpful and constructive comments on our paper. In order to address some of the comments by the reviewers, we have uploaded a revision of the paper, addressing the following:

- Added more background information for readers unfamiliar with the Chess notation as a footnote in the Introduction when we first introduce Figure 1.

- Updated the saliency maps in Figure 7 and Section 3.4 to show the relative saliency (instead of the binarized map as in the original version).

- Expanded the text on gradient-based approaches to saliency maps (non-RL) in Section 4 and their connection to our proposed work.

- Added a discussion section describing the shortcomings of our approach, and of saliency maps  (Section 5).

- Added more details about the experiment setup, in particular, the perturbation technique for each of the evaluations (Appendix A).

- Added examples of the saliency maps for actions that were not the highest Q (in Appendix B), in particular, for the top three moves in a particular chess position.

We are currently carrying out a few more experiments for the next revision:
- Evaluating and analyzing the saliency maps for deep RL agents for Chess.
- Perturbing non-salient pieces for Chess, and evaluating how much the saliency map changes.

---

> ### Author Response · Authors · 2019-11-15
> **Additional revision**
>
> We have uploaded another revision to the paper, here are the salient changes from the last revision:
>
> - Added saliency maps for deep RL agents for Chess in Appendix C.
>
> - Perturbed non-salient parts of the board to observe the effect on the actions, as an approach to evaluate the quality of the saliency map (Section 3.5).
>
> - Added more discussion in Section 5.
>
> - Minor additions to the description of the properties in Section 2.

---

### Public Comment · ~Akanksha_Atrey2 · 2020-05-06
**Discrepancy in SOTA Saliency Map Method**

Interesting work! It is important to develop saliency map methods that capture relevant and specific features. This type of taxonomy is also useful for generating and evaluating explanations in the future. We have a similar paper in ICLR '20 on saliency maps demonstrating this need along with the need to evaluate saliency maps appropriately (https://openreview.net/pdf?id=rkl3m1BFDB). I do have one question. If you compare figure 2 in your paper and Figure 1, 3 and 4 in Greydanus et al. (https://arxiv.org/pdf/1711.00138.pdf), the saliency maps differ by quite a bit despite being on the same environment. I am wondering if there are some hyperparameters that have not been set accordingly which may have lead to these differing maps and could have possibly impacted the user study where visual output of saliency maps is evaluated.

---

> ### Author Response · Authors · 2020-05-15
> **Thanks for the comment**
>
> Hey, thanks for the comment. To generate their maps, we directly used the code they had made available with their paper. We observed that for some frames in Atari their approach generates focused saliency maps, while for others, it generates noisy saliency maps. In contrast, SARFA generates focused saliency maps for most frames. We selected illustrative frames from the gameplay to show this distinction. These look different from the ones in their paper because they chose some other illustrative frames. The full videos for the gameplay showing SARFA Saliency are uploaded to our repository along with the full videos generated by Greydanus et al.

---

### Decision · Program_Chairs · 2019-12-19

**Decision:**

Accept (Poster)

**Comment:**

A new method of calculating saliency maps for deep networks trained through RL (for example to play games) is presented. The method is aimed at explaining why moves were taken by showing which salient features influenced the move, and seems to work well based on experiments with Chess, Go, and several Atari games.

Reviewer 2 had a number of questions related to the performance of the method under various conditions, and these were answered satisfactorily by the reviewers.

This is a solid paper with good reasoning and results, though perhaps not super novel, as the basic idea of explaining policies with saliency is not new. It should be accepted for poster presentation.